# Volume Feature Rendering for Fast Neural Radiance Field Reconstruction

**Kang Han**     **Wei Xiang**[*]     **Lu Yu**
School of Computing, Engineering and Mathematical Sciences
La Trobe University
{k.han, w.xiang, l.yu}@latrobe.edu.au

## Abstract

Neural radiance fields (NeRFs) are able to synthesize realistic novel views from multi-view images captured from distinct positions and perspectives. In NeRF's rendering pipeline, neural networks are used to represent a scene independently or transform queried learnable feature vector of a point to the expected color or density. With the aid of geometry guides either in the form of occupancy grids or proposal networks, the number of color neural network evaluations can be reduced from hundreds to dozens in the standard volume rendering framework. However, many evaluations of the color neural network are still a bottleneck for fast NeRF reconstruction. This paper revisits volume feature rendering (VFR) for the purpose of fast NeRF reconstruction. The VFR integrates the queried feature vectors of a ray into one feature vector, which is then transformed to the final pixel color by a color neural network. This fundamental change to the standard volume rendering framework requires only one single color neural network evaluation to render a pixel, which substantially lowers the high computational complexity of the rendering framework attributed to a large number of color neural network evaluations. Consequently, we can use a comparably larger color neural network to achieve a better rendering quality while maintaining the same training and rendering time costs. This approach achieves the state-of-the-art rendering quality on both synthetic and real-world datasets while requiring less training time compared with existing methods.

## 1 Introduction

For the task of view synthesis, unobserved views of a scene are synthesized from captured multi-view images. This is a long-standing problem that has been studied for several decades in computer vision and computer graphics. The most popular solution to this problem at present is the neural radiance field (NeRF) [18] since it can render views with high fidelity. NeRF represents the density and color of a spatial point in a given direction using a neural network (NN), typically the multilayer perceptron (MLP). With the predicted densities and colors of sampled points along a ray, the NeRF and its variants [2, 3, 28] use the volume rendering technique to aggregate colors to obtain the final rendered pixel color. As many samples are required to render one pixel, the underlying MLP needs to run many times, leading to high computational complexities for both training and rendering. Alternatively, researchers introduced extra learnable features in the form of 3D grids [25, 17], hash tables [19] and decomposed tensors [6, 12, 7, 23] in addition to MLP's parameters to represent a scene's density and color fields. As querying the feature vector of a sampled point by interpolation is much faster than one MLP evaluation, only a small MLP is used to transform the queried feature vector to the density or color, leading to substantial speed acceleration compared with pure MLP representations.

---

[*]Corresponding author.

37th Conference on Neural Information Processing Systems (NeurIPS 2023).

Furthermore, modeling the density field independently by computational efficient representations can greatly reduce the number of samples in the color field. The density representations could be occupancy grids [19], hash grids with less parameters [16], and small MLPs [3]. These coarse density fields enable importance sampling to make the model focus on non-empty points near the surface. As a result, only dozens of samples in the color field (typically represented by both learnable features and NNs) are needed to render a pixel. For example, Zip-NeRF [4] uses 32 samples after importance sampling through efficient density representations.

Nevertheless, standard volume rendering techniques still need to run a color NN dozens of times to render a single pixel. This is why Instant-NGP [19] uses a small color MLP for fast training and rendering, and why the latest Zip-NeRF [4] is comparably slow when using large color MLPs to achieve a better rendering quality. Although a radiance field can be implemented even without NNs [9], the importance of large color NNs for high-quality rendering is undoubted, considering that the state-of-the-art rendering quality can be achieved by pure MLP representations. The authors of Zip-NeRF [4] and NRFF [12] also highlight the importance of large color NNs in modeling view-dependent effects. Unfortunately, the standard volume rendering technique limits the size of color NNs for the sake of fast training and rendering.

One the other hand, research in generative NeRF models [5, 11, 21] and fast NeRF rendering [13] shows that the feature vectors of samples along a ray can be rendered or integrated first to enable one single evaluation of following NNs. However, it is unclear whether this volume feature rendering (VFR) technique can also be used for fast NeRF reconstruction. Towards this end, this paper revisits the VFR method for the purpose of fast NeRF reconstruction. Our investigation reveals that the VFR can be integrated into the training phase of NeRF models, thereby enabling the utilization of large color neural networks to either enhance rendering quality while maintaining the same training time, or to achieve a similar rendering quality with much faster training.

## 2 Related work

The volume rendering technique integrates the colors of samples along a ray according to their densities [15]. A global density field that is differentiable is very effective in finding the underlying geometry of a scene from multi-view 2D color observations using volume rendering. This capability is one of the essential reasons to NeRF's success. As every point in a scene has a density value, the density field is capable of modeling complex geometry. However, standard volume rendering employed by NeRF and its successors [18, 2, 3, 26, 35, 27, 6, 34] suffer from high computational complexity caused by many NN calls even with importance sampling guided by coarse density fields, as discussed in the preceding section.

Representing the underlying geometry by the signed distance function (SDF) leads to a well-defined surface to to enable one sample-based fast rendering [33, 8, 29, 20, 10, 32]. However, the SDF struggles with modeling complex geometry in the context of neural inverse rendering, e.g., trees and leaves, and this is why the best rendering quality is still achieved by volume representations. In this paper, we revisit the VFR framework that uses density as the geometry representation but shares a similar behavior as the SDF that processes a feature vector by color NNs only once. Thus, the VFR inherits the advantage of volume representation in modeling complex geometry and also has the strength of the SDF in single NN evaluation.

Generative NeRF models such as EG3D [5], StyleNeRF [11] and StyleSDF [21] focus on 3D consistency and fidelity of generated content using generalizable neural networks with NeRF's structure. In comparison, 3D reconstruction from multi-view images using NeRF is a different task from 3D generation. As per-scene optimization is required for NeRF reconstruction, the reconstruction speed is a critical issue in this field, but this problem has not been investigated in the existing generative NeRF works. This paper revisits feature integration in volume rendering for fast reconstruction and demonstrate its effectiveness in the NeRF reconstruction task.

It is noted that feature accumulation is also discussed in the Sparse Neural Radiance Grid (SNeRG) [13] mainly for the purpose of modeling specular effects. However, the diffuse color is still obtained using the standard volume rendering method in [13], which requires many times of MLP evaluations. Besides, the SNeRG needs to train a NeRF first (require days to train) and then bakes the trained NeRF to a sparse grid with specular features. In this work, we demonstrate that this pre-training is not necessary, and the accumulated feature vector can be used to predict the final view-dependent

color instead of only the specular color. Our VFR method can be directly optimized from scratch instead of baking the optimized features in the SNeRG. As a result, our method requires significantly less training time but achieves a much better rendering quality compared with the SNeRG.

Integrating feature vectors of samples along the ray is natural in transformer models [24, 30]. For example, Suhail *et al.* proposed an epipolar transformer to aggregate features extracted from reference views on epipolar lines [24]. However, the computational cost of transformer is much higher than MLP and that transformer-based method is significantly (8 times) slower than MLP-based Mip-NeRF [2] on the same hardware. In this work, we demonstrate that the VFR's aggregation only involves a weighted combination of feature vectors based on densities such that the VFR is significantly faster than transformer-based feature aggregation.

## 3 Model architecture

### 3.1 Standard volume rendering

Standard volume rendering integrates color along a cast ray according to density [15]. A high density of a point on the ray indicates a high probability of hitting the surface. The NeRF and its various variants adopt this standard volume rendering technique, where the color and density of a point are typically predicted by a neural network. For a cast ray $\mathbf{r}(t) = \mathbf{o} + t\mathbf{d}$, where $\mathbf{o}$ is the camera origin and $\mathbf{d}$ is the view direction, the rendered color $C(\mathbf{r})$ using standard volume rendering is:

$$C(\mathbf{r}) = \int T(t)\sigma(\mathbf{r}(t))\mathbf{c}(\mathbf{r}(t), \mathbf{d})\, dt, \text{ where } T(t) = \exp\left(-\int_0^t \sigma(\mathbf{r}(s))\, ds\right) \tag{1}$$

where $\sigma(\mathbf{x})$ and $\mathbf{c}(\mathbf{x}, \mathbf{d})$ are the density and color at position $\mathbf{x}$, repsectively. In practice, the above integral is solved by sampling and integration along the cast ray. After transforming the densities to weights by $w(t) = T(t)\sigma(\mathbf{r}(t))$, the color will be rendered as follows:

$$C(\mathbf{r}) = \sum_{i=1}^{N} w_i \text{NeuralNet}\left(F(\mathbf{x}_i), \mathbf{d}\right) \tag{2}$$

where $\mathbf{x}$ is the position of a sample point on the ray and $F$ is a function to query the corresponding feature vector of $\mathbf{x}$. Pure MLP-based methods including the NeRF [18], Mip-NeRF [2], Ref-NeRF [28] and Mip-NeRF 360 [3] represent $F$ by using MLPs. However, this representation imposes a computational burden as $N$ MLP evaluations are required to render a ray. Alternatively, recent research [19, 25, 6] shows that modeling $F$ by extra learnable features is significantly faster than pure-MLP based representations, because linear interpolation of learnable features is much more efficient than MLP evaluations. These learnable features are typically organized in the form of the 3D grids, hash tables, and decomposed tensors. With the learnable features, a small NN can achieve the state-of-the-art rendering quality but accelerate both the training and rendering time significantly.

Furthermore, hierarchical importance sampling guided by coarse geometries is employed to greatly reduce the number of samples in (2). This importance sampling can be achieved by modeling the density fields independently [25, 6, 12], occupancy grid [19], and proposal networks [3, 4]. As shown in Fig. 1, importance sampling makes the model focus on valuable samples on the surface. For instance, by using two levels of importance sampling, Zip-NeRF [4] reduces the final number of valuable samples to 32. However, the color NN still needs to run 32 times to deliver the best rendering quality. Although integrating colors predicted by the color NN is in line with the concept in classical volume rendering, the many color NN calls are the main obstacle for fast training and rendering. On the other hand, the importance of large color NNs for realistic rendering especially in modeling the view-dependent effect is highlighted in recent works [3, 12]. Considering the fact that valuable samples on the surface are close in terms of spatial position (see red sampling points in Fig. 1), the queried feature vectors are also very similar as they are interpolated according to their position. As such, evaluating the color NN with similar input feature vectors is not necessary. In the next subsection, we will introduce the volume feature rendering method that integrates these feature vectors to enable a single evaluation of the color NN.

### 3.2 Volume feature rendering

As shown in Fig. 1, the volume feature rendering method integrates queried feature vectors instead of predicted colors to form a rendered feature vector. A subsequent color NN is then applied to

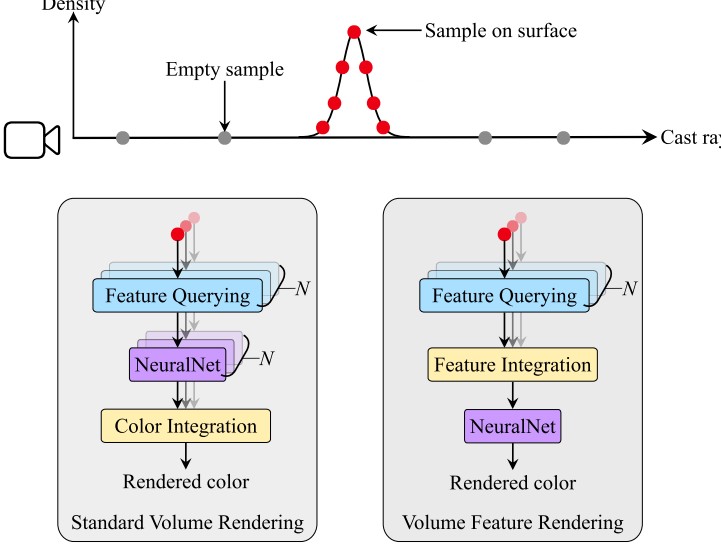

Figure 1: Standard volume rendering needs to run NeuralNet many times for all the samples on the surface, while the volume feature rendering only needs to evaluate NeuralNet once.

transform the feature vector to the final rendered color. By doing this, the color NN only needs to run once. Specifically, we take the NN out of the integral as follows:

$$C(\mathbf{r}) = \text{NeuralNet}\left(\left(\sum_{i=1}^{N} w_i F(\mathbf{x}_i)\right), \mathbf{d}\right). \tag{3}$$

This fundamental change to the standard volume rendering framework relieves the rendering framework from the high computational cost caused by many NN evaluations. Consequently, we can use a larger network to achieve a better rendering quality while maintaining a similar training time.

### 3.3 Pilot network

It is reasonable to integrate feature vectors of non-empty samples in the VFR. However, at the beginning of training, the model does not have a coarse geometry to focus on samples on the surface. The VFR will integrate feature vectors of all samples during the early training stage. We found that

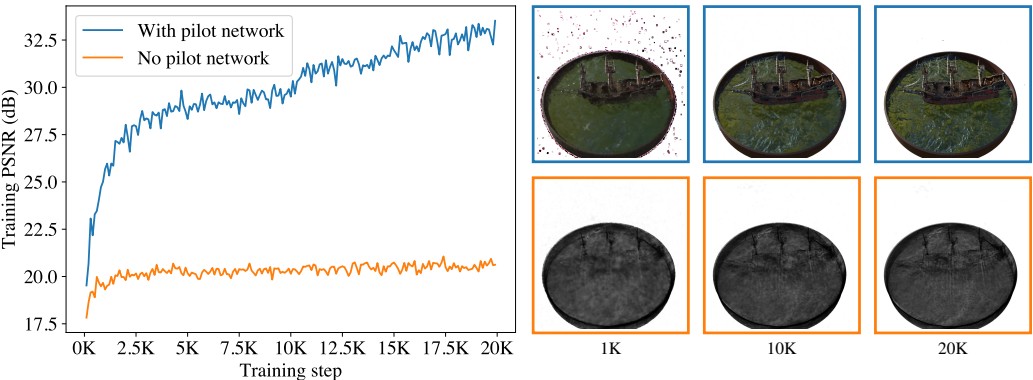

Figure 2: The VFR fails to converge on the scene of *ship* from the dataset in [18]. A small pilot network functioning as standard volume rendering by a few training steps in the early training stage resolves this problem.

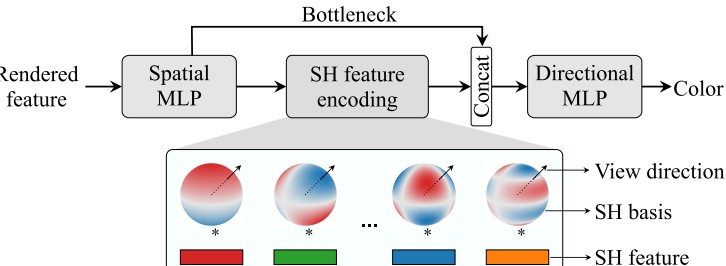

Figure 3: Architecture of the designed neural network for modeling view-dependent effects. Each spherical harmonics basis function is multiplied by a feature vector to encode the input view direction.

this feature integration works well for most scenes but fails to converge for some scenes. Such failure is more likely to occur when employing a comparably large NN. The VFR normally converges using the NN with two hidden layers each of 256 neurons, but it does not converge when using four hidden layers. Fig. 2 illustrates a failure case on the scene of *ship* from the NeRF synthetic dataset [18].

As increasing the size of the NN only has a slight implication to reconstruction and rendering speeds but can lead to a better rendering quality thanks to the VFR framework, we are motivated to use a large NN. We hypothesize that the reason for the aforementioned convergence failure is that the model cannot find the correct geometry due to the feature integration of all samples in the early training stage. Motivated by this insight, we design a pilot network that functions as standard volume rendering to aid the model in finding a coarse geometry in the early training stage. A small pilot network and a few training steps are sufficient to achieve this goal. In this work, we use the pilot network with two hidden layers, each with 64 neurons. The pilot network is only used in the first 300 training steps. After this training, the pilot network will be discarded, and we can use the VFR to train the model normally. The number of pilot training steps is small compared with the total training steps, e.g., 20K steps. Thus, it has a negligible effect on the overall training time. As shown in Fig. 2, the model converges normally with the aid of the pilot network.

### 3.4   View-dependent effect

We use the spherical harmonics (SH) feature encoding method to encode view direction for modeling view-dependent effects. This approach can be seen as a variant of the rendering equation encoding in [12]. Different from the vanilla SH encoding [19] employed in the literature, we multiply each encoded SH coefficient by a feature vector as shown in Fig. 3. Specifically, the rendered feature vector by the VFR is fed into a spatial MLP to produce two feature vectors: SH feature vector and bottleneck feature vector. The SH feature vector will be split into small feature vectors $\mathbf{f}$ for SH coefficients, i.e., $\{\mathbf{f}_0^0 | l = 0\}$, $\{\mathbf{f}_1^{-1}, \mathbf{f}_1^0, \mathbf{f}_1^1 | l = 1\}$, etc. In the SH feature encoding block, the SH feature vector $\mathbf{f}_m^l$ will be multiplied with SH coefficient $Y_l^m(\mathbf{d})$ with view direction $\mathbf{d}$ to form one SH feature encoding vector $\mathbf{e}_m^l$, written as

$$\mathbf{e}_l^m = \mathbf{f}_l^m Y_l^m(\mathbf{d}) \tag{4}$$

A comphrehensive SH encoding is derived by concatenating all encoding vectors as $\mathbf{E} = \{\mathbf{e}_0^0, \mathbf{e}_1^{-1}, \mathbf{e}_1^0...\}$, which will be further concatenated with the bottleneck feature vector (view direction independent). This concatenated vector is used to predict the final rendered color by a directional MLP. The NN in (3) thus includes the spatial and directional MLPs.

## 4   Implemention details

We implement the proposed VFR using the NerfAcc library [16], which is based on the deep learning framework PyTorch [22]. The learnable features are organized by a multiresolution hash grid (MHG) [19]. This feature representation models the feature function $\bar{F}$ in (3) that accepts the input of the position and outputs a feature vector. The color NN contains a spatial MLP and a directional MLP. The spatial MLP has two layers and the directional MLP has four layers, all with 256 hidden neurons. The size of the bottleneck feature is set to 256. We use the GELU [14] instead of the commonly used ReLU [1] activation function, as we found the GELU results in a slightly better quality.

The density of a sample $\mathbf{x}_i$ is derived from the queried feature vector $F(\mathbf{x}_i)$ by a tiny density mapping layer. On the real-world 360 dataset [3], the mapping layer is a linear transform from the feature vector to the density value. On the NeRF synthetic [18] dataset, we found a tiny network with one hidden layer and 64 neurons yields a slightly better rendering quality. The density mapping layers have negligible impact on the overall training time for both datasets.

We employ a hierarchical importance sampling strategy on both the NeRF synthetic [18] and real-world 360 datasets [3]. The occupancy grid is used for efficient sampling on the NeRF synthetic dataset. For the 360 dataset, we follow the most recent Zip-NeRF [4] that uses two levels of importance sampling. These two density fields are implemented by two small MHGs, where the number of hashing levels is ten and the feature channel in each level is one. The final number of samplings on the radiance field is 32. The models are trained using the Adam optimizer with a learning rate of 0.01 and the default learning rate scheduler in NerfAcc [16]. We use PSNR, SSIM [31] and LPIPS [36] to evaluate the rendered image quality. The running times are measured on one RTX 3090 GPU. Table 1 summarizes other experimental settings in this work.

Table 1: Experimental settings on the NeRF synthetic and real-world 360 datasets. #Features represents the number of learnable features in the underlying hash grids, which is estimated by hash grid hyperparameters (HGH), i.e., the number of levels$\times$ the number of voxels each level$\times$the number of feature channels.

| Dataset | Model | #MHG | HGH | #Batch | SH degree | #SH feature |
|---|---|---|---|---|---|---|
| NeRF synthetic[18] | VFR-small | 13M | $16\times2^{19}\times2$ | $2^{18}$ points | 4 | 4 |
| Real-world 360[3] | VFR-base | 34M | $32\times2^{19}\times2$ | $2^{13}$ rays | 7 | 8 |
| Real-world 360[3] | VFR-large | 134M | $32\times2^{21}\times2$ | $2^{13}$ rays | 7 | 8 |

# 5 Experimental results

## 5.1 NeRF synthetic dataset

The VFR achieves the best rendering quality but uses the minimum training time compared with state-of-the-art fast methods on the NeRF synthetic dataset. Since the original Instant-NGP [19] uses the alpha channel in the training images to perform background augmentation, we report the results of the Instant-NGP from NerfAcc [16], where such augmentation is not employed for fair comparison. As shown in Table 2, the VFR outperforms Instant-NGP by 2.27 dB and NerfAcc by 1.51 dB, but only uses 3.3 minutes for training. Compared with TensoRF [6], our method only uses 33% of its training time while surpassing TensoRF by 1.48 dB in PSNR. In addition, the VFR achieves the best SSIM and LPIPS on this dataset, which is consistent with the PSNR metric.

This significant improvement in rendering quality stems from the increased size of the NN. The MLP in Instant-NGP and NerfAcc has 4 layers and 64 neurons in each layer. By comparison, we use a six-layer MLP with 256 neurons. The computational cost of our MLP is roughly 24 times higher than MLPs in Instant-NGP and NerfAcc. However, thanks to the VFR framework, our MLP only needs to run one time instead of many times as in the standard volume rendering framework employed in the

Table 2: Rendering quality and training time on the NeRF synthetic dataset. The VFR trained with 6K steps in 0.97 minutes achieves a similar quality compared with existing methods.

| | PSNR↑ | SSIM↑ | LPIPS↓ | Training time↓ |
|---|---|---|---|---|
| NeRF [18] | 31.69 | 0.953 | 0.050 | hours |
| Mip-NeRF [2] | 33.09 | 0.961 | 0.043 | hours |
| TensoRF [6] | 33.14 | 0.963 | 0.047 | 10 mins |
| Instant-NGP [19] | 32.35 | 0.960 | 0.042 | 4.2 mins |
| NerfAcc [16] | 33.11 | 0.961 | 0.053 | 4.3 mins |
| VFR-small: 6K | 33.02 | 0.960 | 0.055 | 0.97 mins |
| VFR-small: 20K | **34.62** | **0.971** | **0.038** | **3.3 mins** |

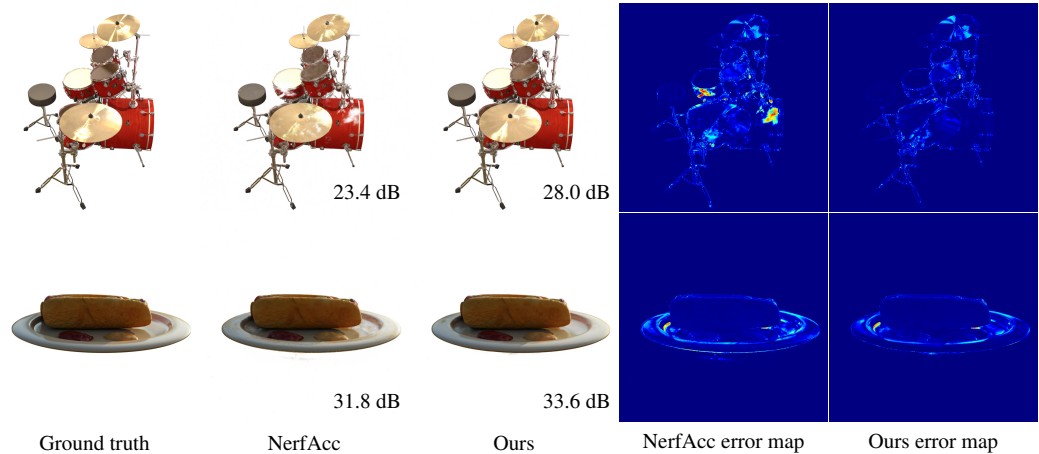

|       | 23.4 dB | 28.0 dB |       |       |
|-------|---------|---------|-------|-------|
|       | 31.8 dB | 33.6 dB |       |       |
| Ground truth | NerfAcc | Ours | NerfAcc error map | Ours error map |

Figure 4: Visual comparison of the synthesized novel views on the scenes of (from top to bottom) *drums* and *hotdog* in the NeRF synthetic dataset.

compared methods. As can be observed from Table 2, the single color NN evaluation property in the VFR leads to a minimum training time even when we use a much larger color NN. Noticeably, the VFR only needs 6K training steps to reach a similar rendering quality compared with the existing algorithms. Such 6K training steps can be completed within 1 minute on one RTX 3090 GPU, which is less than the 25% of training times of Instant-NGP and NerfAcc, and only 10% of the training time in TensoRF.

Fig. 4 provides a visual comparison of the rendered novel views by NerfAcc (based on Instant-NGP) and the VFR. On the compared scenes, both NerfAcc and the VFR can recover good geometries. However, for the scene of *drums*, obvious visual artefacts appear in the rendered views from NerfAcc while our results do not have such visual artefacts. Besides, the VFR produces more realistic visual effects such as shadow and reflection, as observed on the scenes of *hotdog*. The presented error maps and objective PSNRs in Fig. 4 also demonstrate the advantage of the VFR in terms of rendering quality. While NerfAcc and our method use the same feature representation, i.e., hash grids, we believe the quality advantage of our method stems from the large color NN enabled by the VFR.

## 5.2 Real-world 360 dataset

We also experiment on seven real-world unbounded scenes from the 360 dataset [3]. The scenes of *flowers* and *treehill* in this dataset are not included as they are not publicly available. We use the unbounded ray parameterization method proposed in Mip-NeRF 360 [3], where points are mapped to a sphere if their radius to the scene center is larger than one.

The visual comparison in Fig. 5 shows that the VFR is able to render views in similar quality with Zip-NeRF but render more realistic views than NerfAcc (based on Instant-NGP). The VFR provides a more accurate geometry reconstruction than NerfAcc on the scene of *bicycle*. As can be observed from Fig. 5, the rendering results of NerfAcc contain obvious visual artefacts on the scenes of *bonsai* and *kitchen*, while the VFR's results are more realistic and closer to the ground truth. The reflective effect in the highlighted patches on the scene of *kitchen* is also well modeled by our model thanks to our new SH feature encoding method.

As shown in Table 3, the VFR is able to achieve similar rendering quality to the state-of-the-art Zip-NeRF [4] but using the minimum training time. Although Mip-NeRF 360 is able to synthesize novel high-fidelity views, it requires days to train on high-end GPUs because it is based on a pure MLP representation. Using learned features organized in hash grids, Instant-NGP achieves a comparable rendering quality but reduces the training time from days to hours. The most recent Zip-NeRF uses multisampling for high-quality anti-aliasing view rendering at the expense of training time relative to Instant-NGP. The VFR achieves a slightly better rendering quality than Zip-NeRF in terms of the PSNR. However, we achieve this state-of-the-art rendering quality with a significantly less training

Table 3: Rendering quality and training time on the real-world 360 dataset. The scale factor for NerfAcc and the VFR-base models remains constant at four for both outdoor and indoor scenes. For the VFR-large model, the scale factor is adjusted to four for outdoor scenes (*bicycle, garden, stump*) and two for indoor scenes (*bonsai, counter, kitchen, room*) in order to maintain conformity with Zip-NeRF [4].

|  | PSNR↑ | SSIM↑ | LPIPS↓ | #Features↓ | Training time↓ |
|---|---|---|---|---|---|
| NeRF [18] | 24.85 | 0.659 | 0.426 | N/A | days |
| Mip-NeRF [2] | 25.12 | 0.672 | 0.414 | N/A | days |
| NeRF++ [35] | 26.21 | 0.729 | 0.348 | N/A | days |
| Mip-NeRF 360 [3] | 29.11 | 0.846 | 0.203 | N/A | days |
| Instant-NGP [19] | 27.06 | 0.796 | 0.265 | 84M | 0.81 hrs |
| Zip-NeRF [4] | 29.82 | **0.874** | **0.170** | 84M | 5.20 hrs |
| NerfAcc [16] | 28.69 | 0.834 | 0.221 | 34M | 11 mins |
| VFR-base: 20K | 29.48 | 0.830 | 0.233 | 34M | 5.7 mins |
| VFR-base: 40K | 29.92 | 0.850 | 0.208 | 34M | 11 mins |
| VFR-large: 20K | 29.51 | 0.846 | 0.252 | 134M | 11 mins |
| VFR-large: 40K | **29.90** | 0.862 | 0.231 | 134M | 22 mins |

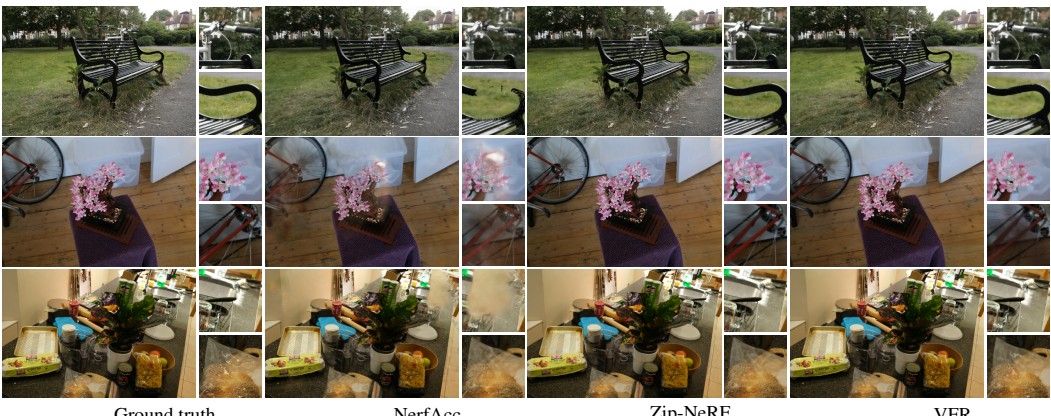

Ground truth         NerfAcc         Zip-NeRF         VFR

Figure 5: Visual comparison of synthesized novel views on the scenes of (from top to bottom) *bicycle*, *bonsai*, and *kitchen* on the real-world 360 dataset [3].

time. As a comparison, the VFR-large model only uses 22 minutes to train on one RTX 3090 GPU, while Zip-NeRF's training time is 5.2 hours on the same GPU.

It should be noted that we achieve this high-quality view rendering using an entirely different approach than Zip-NeRF. As Zip-NeRF focuses on anti-aliasing, multisampling is a reasonable solution but leads to a higher computational cost. Besides, the authors of Zip-NeRF found a larger color NN helps in improving the rendering quality. Since Zip-NeRF still employs the standard volume rendering method, a larger color NN will undoubtedly increase the training time. The VFR, however, relieves the rendering framework from the high computational cost caused by many color network evaluations. As only one color network evaluation is required in the VFR, NN evaluation is no longer the computing bottleneck in the rendering framework. As a result, the VFR enables one to achieve a similar rendering quality as Zip-NeRF but with a significantly less training time.

### 5.3  Ablation study

Table 4 shows the ablation study of the proposed methods. Compared with the commonly used ReLU, we found that GELU activation improves the rendering quality for both the standard volume rendering and the VFR. Larger MLP does increase the quality using the standard volume rendering but at the expense of more training time. For the VFR, when increasing the size of the NN by using more hidden neurons, the rendering quality constantly improves but the training time only slightly

Table 4: Ablation study. SH means spherical harmonics for view direction encoding (VDE) while SHFE stands for SH feature encoding. We use the NerfAcc's implementation of Instant-NGP as a baseline. MLP's size is represented by layers×neurons. Times are measured in minutes by 20K training steps.

| | Activation | MLP | VDE | NeRF synthetic (VFR-small) | | | | Real-world 360 (VFR-base) | | | |
| | | | | Time↓ | PSNR↑ | SSIM↑ | LPIPS↓ | Time↓ | PSNR↑ | SSIM↑ | LPIPS↓ |
|---|---|---|---|---|---|---|---|---|---|---|---|
| Baseline | ReLU | 4×64 | SH | 4.4 | 33.11 | 0.9614 | 0.053 | 5.6 | 28.39 | 0.8147 | 0.245 |
| +GELU | GELU | 4×64 | SH | 4.4 | 33.43 | 0.9634 | 0.049 | 5.6 | 28.43 | 0.8152 | 0.245 |
| +Medium MLP | GELU | 4×128 | SH | 5.6 | 33.92 | 0.9663 | 0.045 | 6.8 | 28.70 | 0.8205 | 0.239 |
| VFR | ReLU | 4×64 | SH | 2.9 | 33.17 | 0.9619 | 0.053 | 5.0 | 28.20 | 0.8013 | 0.262 |
| +GELU | GELU | 4×64 | SH | 3.0 | 33.61 | 0.9641 | 0.048 | 5.0 | 28.26 | 0.8064 | 0.254 |
| +Medium MLP | GELU | 6×64 | SHFE | 3.0 | 34.05 | 0.9678 | 0.043 | 5.2 | 29.06 | 0.8215 | 0.240 |
| +Large MLP | GELU | 6×128 | SHFE | 3.0 | 34.33 | 0.9697 | 0.041 | 5.4 | 29.28 | 0.8266 | 0.236 |
| W/O SHFE | GELU | 6×256 | SH | 3.2 | 34.46 | 0.9701 | 0.040 | 5.5 | 28.91 | 0.8192 | 0.245 |
| Full model | GELU | 6×256 | SHFE | 3.3 | **34.62** | **0.9713** | **0.038** | 5.7 | **29.48** | **0.8301** | **0.233** |

increases. Replacing the SH encoding with the SH feature encoding (W/O SHFE vs Full model) also leads to a better quality with negligible extra training time.

# 6    Limitations

Similar to many other neural rendering methods, the VFR requires per-scene optimization to deliver high-fidelity view rendering. Although the VFR can be optimized in several minutes for one scene, we are still far from realizing real-time reconstruction for 3D video applications. As for the rendering time, our method can render 4∼5 frames per second at the resolution of 800×800 on the NeRF synthetic dataset, similar to the rendering speed of NerfAcc. On the real-world 360 dataset, the rendering speed is about 1∼2 frames per second at the resolution of 1300×840.

Elaborate implementation optimization can help in achieving high-quality real-time rendering using this VFR. However, compared with the recent work that specifically focuses on real-time rendering such as BakedSDF [33] and MobileNeRF [8], the rendering speed of the VFR needs to be further improved. BakedSDF and MobileNeRF represent a scene's surface to enable one feature querying and one neural network evaluation, at the expense of some rendering quality decline compared with volume representation. The VFR uses the volume representation with one color NN evaluation to deliver the state-of-the-art rendering quality, but the feature querying still needs to be conducted many times for a cast ray. Considering the fact that the queried feature vectors of non-empty samples are finally integrated into one feature vector using the VFR, we believe the number of feature querying can also be reduced to a small value, e.g., from 32 to 5, to achieve high-quality and real-time rendering simultaneously. An additional prospective constraint of the VFR lies in its potential inefficacy when confronted with semitransparent objects as it integrates feature vectors first and then predicts a single final color. This property is not well reflected in the NeRF synthetic and real-world 360 datasets and requires further investigation in future work.

# 7    Conclusion

We revisited the volume feature rendering method in the context of fast NeRF reconstruction. The VFR integrates the queried feature vectors of non-empty samples on a ray and then transforms the integrated feature vector to the final rendered color using one single color neural network evaluation. We found the VFR cannot guarantee the convergence on some scenes and thus introduced the pilot network to guide the early-stage training to tackle this problem. The VFR is able to achieve the state-of-the-art rendering quality while using significantly less training time. It has a great potential to replace the standard volume rendering in many neural rendering methods based on volume representation, with the benefits of either a better rendering quality by employing a larger color neural network or a significantly reduced training time thanks to one single color neural network evaluation.

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
