# OpenReview forum: "Volume Feature Rendering for Fast Neural Radiance Field Reconstruction"
_NeurIPS.cc/2023/Conference — NeurIPS 2023 poster_

### Official Review · Reviewer_q1tF · 2023-06-21

**Soundness:** 3 good
**Presentation:** 3 good
**Contribution:** 4 excellent
**Rating:** 9
**Confidence:** 5

**Summary:**

This work improves grid-based NeRF on both quality and training speed. To predict view-dependent color, they proposes to condition MLP on the volume rendered voxel features instead of each original point features. As a result, the MLP only run once for each pixel instead of the original dozen of MLP evaluations. To improve robustness, they find using the old pipeline (called pilot network) to warmup the scene for few hundreds iteration is enough. A new spherical-harmonic-based view-direction feature encoding is proposed, which combined with larger MLP achieve state-of-the-art quality. Experiments results show the superior quality and training speed from previous arts.

**Strengths:**

The improvement is really solid and promising. The proposed method is not hard to adapt to other method. The paper is easy to follow as well. I believe this work can benefit many future researches.

**Weaknesses:**

It's a pity that videos are not provided to showcase the improvement on view-dependent effect.

**Questions:**

1. **Eq.4.** Is the final view-direction feature summed toghether ($\sum_{l,m} e_l^m$) like what we conventionally do for rgb or concatenated ($\mathrm{concant}(\{e_l^m | \forall l,m\})$)?

2. **Table 3.** It's better to put a table footnote that some baselines' training times are rescaled by 8 (the number of GPUs they used). As discussed in L216-219, this is not rigorous so I think it's better to inform readers who only skims the table.

3. **Table 4's H row.** It seems that the 56 should have be 256.


**Limitations:**

I think the limitation is well discussed.

---

> ### Author Rebuttal · Authors · 2023-08-09
>
> Thanks for your encouraging feedback. We respond to your concerns one-by-one as follows.
>
> **Weaknesses:**
>
> __Videos are not provided:__ Thanks for your suggestion. We will upload the rendered videos to showcase the improvement on view-dependent effect in our final version of this paper.
>
> **Questions:**
> 1. __Eq. 4:__ The encoded view direction feature vectors are concatenated to form one comprehensive view direction encoding. We will provide more detailed description for SH feature encoding to highlight this feature vector concatenation in our final version.
>
> 2. __Table 3:__ Thanks for your valuable suggestion. We provide the running time comparison of various methods measured on the same RTX 3090 for fair and rigorous comparison. As shown in the updated Tables 1 and 2, we can draw the same conclusion as in our first submission.
>
> 3. __Table 4’s H row:__ Thanks for pointing this out. We will correct this error in our final version.
>
> Table 2. PSNR and training time on the NeRF synthetic dataset on one RTX 3090.
> |       | PSNR | Time
> | ------ | ------ | --- |
> | NeRF          | 31.69   | hours
> | Mip-NeRF   | 33.09   | hours
> | TensoRF      |  33.14  | 10 mins
> | Instant-NGP | 32.35  | 4.2 mins
> | NerfAcc       |  33.11  | 4.2 mins
> | Our VFR: 6K | 33.02 | 0.97 mins
> | Our VFR: 20K | __34.62__ | __3.3 mins__
>
>
> Table 3. PSNR and training time on the real-world 360 dataset on one RTX 3090.
> |       | PSNR | #Feature | Time
> | ------ | ------ | --- | --- |
> | NeRF          | 24.85   | N/A | days
> | Mip-NeRF   | 25.12   | N/A  |days
> | NeRF++      |  26.21  | N/A |days
> | Mip-NeRF360 | 29.11  | N/A |days
> | Instant-NGP       |  27.06  | 84M |0.81 hrs
> | Zip-NeRF | 29.82 | 84M | 5.2 hrs
> |NerfAcc | 28.69 | 34M | 11 mins
> |Our VFR: 20K| 29.48 | __34M__ | __5.7 mins__
> |Our VFR: 40K| __29.92__ | __34M__ | __11 mins__

---

> > ### Comment · Reviewer_q1tF · 2023-08-19
> >
> > I appreciate the author's responses. I don't have any other questions.

---

### Official Review · Reviewer_y3xd · 2023-07-05

**Soundness:** 3 good
**Presentation:** 3 good
**Contribution:** 1 poor
**Rating:** 4
**Confidence:** 5

**Summary:**

The paper proposes a method for fast NeRF reconstruction. The main contribution of the paper
is that instead of integrating radiance along the camera ray which requires evaluation of the
color MLP at each point, the paper proposes to integrate features along the ray and only
evaluate a single MLP on the integrated features to get the final pixel color. In this way,
the proposed method can reduce the number of MLP evaluations, thus allowing the method to
use achieve faster reconstruction even with a larger MLP. The larger MLP also allows the method
to achieve higher-quality reconstructions. The paper does experiments on NeRF synthetic dataset
and real-world 360 dataset and demonstrates higher accuracy and fast reconstruction speed
than baseline methods such as instant-nap, nerfacc, and tensorf.



**Strengths:**

1. The paper shows that integrating the features along the ray and decoding them to the
pixel color with a large MLP can achieve faster and higher-quality reconstructions.

2. The paper proposes to use a small pilot network at the beginning of the optimization
that applies standard volume rendering, which helps stabilize the training of the proposed method.

3. The paper did a thorough evaluation of the proposed method against different baseline methods
on both synthetic and real-world datasets.


**Weaknesses:**

See Questions.

**Questions:**

1. It's not clear to me how the method integrates the features along the ray. In Equation 3,
the paper says it performs a weighted combination of the features. I would assume that the
weight is calculated in a similar way to the standard volume rendering based on the density
field. Then it's not clear to me how the density field is predicted. The paper is based on
the NGP representation, and it in fact uses a MLP to predict the density of each point. Is
the paper doing in the same way? If that's the case, I think the paper should make it clear.
Also in this case, the paper should not claim that it is performing a single evaluation NN
for each pixel (or make it more clear, it's a single evaluation of color MLP).
Similarly, in the case of real-world scenes, the paper uses proposal networks for sampling,
which also requires additional MLP evaluations. Overall, I think the paper should make it
more adequate when it says it's performing a single neural network evaluation.

2. In Line 70, the paper says SNeRG still requires many times of MLP evaluations to get
the diffuse color. This is not true. SNeRG stores the diffuse colors at each voxel, and
directly applies alpha compositing to get the diffuse color for the ray. Therefore, no
MLP evaluation is needed.

3. The motivation behind the pilot network is kind of ad-hoc to me. First, the paper
says VFR works well for most scenes but fails to converge for some scenes. If it's a scene-specific
problem, It's good to know the characteristics of the scenes where it will fail.  In addition,
the paper says the reason for VFR not converging is that it does not have a coarse geometry to focus on samples on
the surface. The first question is that why VFR needs to focus on samples on the surface. Does it
mean that VFR will not work well on objects that don't have an opaque surface (like furry objects)?
Why standard volume rendering does not this problem? Moreover, the paper also says larger networks
tend to have more convergence issues than smaller networks, how does this relate to the lack of
coarse geometry? I think the paper should have a better explanation on the convergence issues
with VFR and make a more convincing argument behind the motivation of the pilot network.

4. In Figure 4, the results on the Ficus scene of Nerfacc is suspicious to me. What's the reason
for the weird specularity on the pot?

5. I do have concerns over the technical contributions of the paper. As the paper said in Line 68,
SNeRG has been using feature integrations for faster NeRF rendering. In addition, previous works such as StyleNeRF, EG3D, and StyleSDF have also been using feature integration for NeRF. While
they don't do a systematic evaluation on the per-scene reconstruction task, I think generally
feature integration has been a standard technique in the NeRF community, and has been explored
in similar tasks as mentioned above.

To summarize, while I appreciate the thorough experiments and evaluations of the paper, my major
concern over the paper is that the proposed method has been explored in similar tasks in
previous works and there is a lack of technical contributions from the paper.


**Limitations:**

The limitations look good to me.

---

> ### Author Rebuttal · Authors · 2023-08-09
>
> We appreciate your insightful comments for our manuscript. We respond to your concerns one-by-one as follows.
>
> 1. __How the method integrates the features along the ray:__ We apologize for the missed details of how to obtain the integration weights. The weights are calculated from densities of sampled points, where density of a point is predicted by a linear layer from the queried feature vector before integration. As this linear layer only has one output neuron to predict the density value, it has a negligible effect on the overall training time. On the real-world 360 scenes, the proposal networks do require additional MLP evaluations for density prediction as mentioned by the reviewer. However, these MLPs for the density field are usually very small and MLPs are even not necessary like that in TensoRF. Thus, MLP evaluations for the density field have a very small effect on the overall training time. The reduction in the number of MLP evaluations for color prediction is more valuable as the color MLP evaluation is typically one of the bottlenecks for the overall training time. Nevertheless, we appreciate the reviewer’s suggestion in making the argument clear and will adopt the suggestion by revising the argument on a single evaluation of color MLP in our final version.
>
> 2. __The SNeRG still requires many times of MLP evaluations:__ This statement is true because the SNeRG needs to obtain the diffuse color through many MLP evaluations first and then storing the obtained diffuse color for fast rendering. In inverse rendering, the diffuse color of each voxel is unknown and needs to be estimated in NeRF. Thus, the training of the SNeRG is identical to the original NeRF using the standard volume rendering. We agree with the reviewer that no MLP evaluation is needed for SNeRG in the rendering stage. But the SNeRG needs to train a standard NeRF first and then bakes the trained NeRF to a sparse grid. This training and baking in the SNeRG need many MLP evaluations. In our work, we show that the many times of MLP evaluations in the training stage can be avoided by our VFR, leading to a significant reduction in training time for our method (several minutes) and a better rendering quality than the SNeRG (days to train).
>
> 3. __The motivation behind the pilot network:__
>     * We found that our VFR without the pilot network is more likely to fail when the scene is of complex geometry.
>     * Our VFR needs to focus on samples on the surface because the feature vectors of these samples will be integrated into one feature vector to predict the color of that surface point. If the feature vectors from the foreground and background are fused together (could happen when there is no coarse geometry in the early training stage), the following MLP may have difficulties in predicting the correct color. For objects that do not have an opaque surface, our VFR may not work well, and this potential limitation needs to be further investigated in our future work. The standard volume rendering does not have the convergence issue because the color prediction of each sample by the MLP is independent.
>     * We believe larger networks tend to have more convergence issues in our model because larger networks have stronger learning ability, and they try to directly predict the correct color from the integrated feature vector of all samples on a ray and completely ignore the scene geometry. As the training of NeRF models is to find the correct geometry and the color field simultaneously, the above ignorance of geometry leads to failed convergence.
>     * We will add the above explanations to our final version.
>
> 4. __Weird specularity on the pot in Fig 4:__ We believe that specularity on the pot is caused by insufficient modeling of the view-dependent effect using a comparably small MLP in the NerfAcc (based on Instand-NGP).
>
> 5. __Technical contributions of the paper:__ As aforementioned in our response to your question on SNeRG, the SNeRG aims at faster NeRF rendering without considering the training time. The SNeRG still needs to evaluate the MLP for many times in the training stage, while our VFR can directly integrate queried feature vectors to enable one large MLP evaluation in the training stage, leading to a significant faster training speed as well as a better rendering quality than the SNeRG. We agree with the reviewer that some works also employ a similar idea from different perspectives. But none of existing works investigate the direct integration of queried feature vectors as ours to accelerate the training of NeRF-like models and to improve the rendering quality for the per-scene reconstruction task. The mentioned works (i.e., StyleNeRF, EG3D, and StyleSDF) are in the field of 3D generation instead of neural inverse rendering from multiple views. Although these works employ feature integration, they all require one MLP (large or small in size) evaluation first for each sampled point and then integrate the yielded feature vectors from the MLP evaluations.
>
>     We respectfully disagree with the reviewer that feature integration has been a standard technique in the NeRF community. We believe that it is not a common view among the NeRF community that queried feature vectors from feature representations like multi-resolution hash encoding can be directly integrated to enable one single-time MLP evaluation. This point is evidenced by the fact that many of the latest representative NeRF works are still based on the standard volume rendering (such as Instant-NGP, TensoRF, and the concurrent Zip-NeRF), although feature integration offers significant benefits as identified by our work. This paper formally introduces the volume feature rendering for per-scene reconstruction task, identifies the potential problem and proposes the solution, conducts extensive experiments and provides a detailed analysis. As such, we believe the contributions of this work are solid and the proposed methods are valuable to the NeRF community.

---

> > ### Comment · Reviewer_y3xd · 2023-08-20
> > **Reply to the authors**
> >
> > I thank the authors for the rebuttal.
> >
> > While I agree that feature integration is not fully explored in the multi-view Nerf reconstruction task, still I want to point out that it's a technique that has been widely used in the generative NeRF tasks such as EG3D and StyleNeRF. The real contribution of the paper is to re-study it in the context of NeRF reconstruction.  However, in the current format, the paper is claiming it's introducing a new method (which is not) for volume rendering without referring to previous works mentioned above. I believe that the paper should be rewritten to better discuss the relationship with previous works and state why it's non-trivial to adapt such a technique to the NeRF reconstruction task.

---

> > > ### Comment · Reviewer_WTsx · 2023-08-20
> > >
> > > In the NeRF real-time rendering community, feature integration is also there for quite a while; see Fig. 2 of this work: Baking Neural Radiance Fields for Real-Time View Synthesis. Hence I kind of agree that the real contribution of this work is to adapt such technique to reconstruction of NeRF.

---

> > > > ### Author Response · Authors · 2023-08-21
> > > >
> > > > Thanks for your comment. We agree with you that feature integration is used for real-time rendering in the mentioned SNeRG. As we discussed in the related work, the training of SNeRG is based on the standard volume rendering that requires many MLP calls. In comparison, feature integration investigated in this work enables one single MLP call in the training stage, leading to significantly less training time and better rendering quality than SNeRG.  We will revise the contribution claim according to your comment to highlight our contribution in reconstruction of NeRF in the final version of this paper.

---

> > > ### Author Response · Authors · 2023-08-21
> > >
> > > Thanks for your valuable comments. We agree with you that the real contribution of the paper is to re-study feature integration in the context of NeRF reconstruction. However, we believe this contribution is still solid, considering that this work has achieved a significant reconstruction speed improvement compared with the state-of-the-art works on the NeRF reconstruction task.
> > >
> > > We will take your advice by revising the main contribution of this paper from proposing a new method to re-studying the feature integration in the field of multi-view NeRF reconstruction in the final version. In addition, we will also follow your comment to discuss the relationship with previous works and elaborating on why it is non-trivial to adapt such a technique to the NeRF reconstruction task.
> > >
> > > Specifically, generative NeRF models such as EG3D, StyleNeRF and StyleSDF also employ feature integration for the 3D scene generation task. However, these works focus on 3D consistency and fidelity of generated content using generalizable neural networks with NeRF’s structure. In comparison, 3D reconstruction from multi-view images using NeRF is a very different task from 3D generation. As per-scene optimization is required for NeRF reconstruction, the reconstruction speed is a critical issue in this field, but this problem has not been investigated in the existing generative NeRF works. This paper studies feature integration in volume rendering for fast reconstruction and demonstrate its effectiveness in the NeRF reconstruction task.
> > >
> > > Furthermore, adapting feature integration to the NeRF reconstruction task is non-trivial because the model may fail to converge due to the lack of coarse geometry at the early training stage. This paper is the first to find this convergence problem as feature integration has not been studied in the NeRF reconstruction task. Based on this finding, we propose a pilot network to solve this problem. In summary, we believe the comprehensive re-study of feature integration in the field NeRF reconstruction conducted in this work makes a solid contribution to the NeRF community.

---

### Official Review · Reviewer_WTsx · 2023-07-06

**Soundness:** 2 fair
**Presentation:** 3 good
**Contribution:** 2 fair
**Rating:** 5
**Confidence:** 5

**Summary:**

The authors propose to perform feature accumulation first at each pixel location, and then pass the accumulated feature to a MLP to get the rendered color in NeRF. They show improved rendering quality over baseline methods on NeRF synthetic dataset and object-centric 360-degree captures.

**Strengths:**

1. Idea seems simple and easy to reproduce.
2. Writing is good and paper is easy to follow overall.
3. Evaluation seems extensive, though some important aspects might be missing (see bullet point 1, 3 in the weakness section).

**Weaknesses:**

1. Will jittering happen when rendering videos using the proposed method? I feel the proposed method might be more likely to suffer from jittering issues than standard volume rendering, as the large MLP network might introduce view-consistencies. It would be great to include some video results in the demo.

2. Training time improvement doesn't seem to be that big, compared with NeRFACC that the proposed method is built upon. Is it due that volume-rendering of a relatively high-dimensional features is slower than volume-rendering of RGB colors, hence cutting down the performance gain from reducing MLP evaluations?

3. Table 2 and Table 3 only contains PSNR comparisons against baselines. I don't think this is indicative enough of image sharpness; I would like to see SSIM and LPIPS comparisons to make a better judgement.

**Questions:**

See bullet point 1, 2, 3 in the weakness section.

**Limitations:**

The limitations seem to be discussed in detail in the paper.

---

> ### Author Rebuttal · Authors · 2023-08-09
>
> Thanks for your valuable comments on our manuscript. We address your concerns as follows.
>
> 1. __Jittering problem:__ We understand jittering is the problem of 3D inconsistency when rendering video with changing viewpoints. We do not observe the consistency difference between our rendered videos and those obtained by using standard volume rendering methods. As MLPs are conditioning on the queried feature vectors for both ours and standard volume rendering methods, a larger MLP does not introduce extra 3D inconsistency. In addition, considering that pure large MLP-based representations (e.g., the original NeRF and mip-NeRF) are able to render videos with high 3D consistency, we believe large MLPs might not be a source of jittering issue.  We will add rendered videos in the final version to demonstrate 3D consistency.
>
> 2. __Training time improvement:__ Volume rendering of relatively high-dimensional features (e.g., 64 channels) instead of RGB colors has a negligible effect on the overall training time. Two main computations in the rendering process are feature querying and MLP evaluation, so the overall training time is not only dependent on the number of MLP evaluations. As shown in Table 2, to reach a similar quality, our method is 4+ times faster than the NerfAcc based on the standard volume rendering on the NeRF synthetic dataset. On the real-world 360 dataset, our method achieves a 0.79 dB improvement in PSNR while only taking 52% of its training time, compared with NerfAcc. As NerfAcc based on Instant-NGP is the most state-of-the-art method in fast training, we believe our training time improvement is significant and our method is valuable to the community.
>
> 3. __SSIM and LPIPS comparisons:__ Thanks for your suggestion. We add SSIM and LPIPS results in Tables 2 and 3 as follows. On the NeRF synthetic dataset in Table 2, our method outperforms the compared methods in all metrics including PSNR, SSIM and LPIPS while using the minimal training time. In Table 3, it is observed that our method achieves similar rendering quality (slightly better PSNR but slightly worse SSIM and LPIPS) as the concurrent Zip-NeRF on the real-world 360 dataset. However, our method requires significantly less parameters and less training time (11 minutes vs 5.2 hours) compared with Zip-NeRF. In addition, our method only uses 52% of NerfAcc’s training time to reach better PSNR and similar SSIM and LPIPS compared with NerfAcc.
>
> Table 2. Rendering quality and training time on the NeRF synthetic dataset on one RTX 3090.
> |       | PSNR | SSIM | LPIPS | Time
> | --- | --- | --- | --- | --- |
> | NeRF          | 31.69   | 0.953 | 0.050 | hours
> | Mip-NeRF   | 33.09   | 0.961 | 0.043 | hours
> | TensoRF      |  33.14  | 0.963 | 0.047 | 10 mins
> | Instant-NGP | 32.35  | 0.960 | 0.042 | 4.2 mins
> | NerfAcc       |  33.11  | 0.961 | 0.053 | 4.2 mins
> | Our VFR: 6K | 33.02 | 0.960 | 0.055 | 0.97 mins
> | Our VFR: 20K | __34.62__ |  __0.971__ | __0.038__ | __3.3 mins__
>
>
> Table 3. Rendering quality and training time on the real-world 360 dataset on one RTX 3090.
> |       | PSNR | SSIM | LPIPS | #Feature | Time
> | --- | --- | --- | --- | --- | --- |
> | NeRF          | 24.85   | 0.659 | 0.426 | N/A | days
> | Mip-NeRF   | 25.12   | 0.672 | 0.414 | N/A  |days
> | NeRF++      |  26.21  | 0.729 | 0.348 | N/A |days
> | Mip-NeRF 360 | 29.11  | 0.846 | 0.203 | N/A |days
> | Instant-NGP       |  27.06  | 0.796 | 0.265 | 84M |0.81 hrs
> | Zip-NeRF | 29.82 | __0.874__ | __0.170__ | 84M | 5.2 hrs
> |NerfAcc | 28.69 | 0.834 | 0.221 | 34M |11 mins
> |Our VFR: 20K| 29.48 | 0.830 | 0.233 | __34M__  | __5.7 mins__
> |Our VFR: 40K| __29.92__ | 0.850 | 0.208 | __34M__ | __11 mins__

---

> > ### Comment · Reviewer_WTsx · 2023-08-20
> >
> > Thanks for the response. I have no other questions.

---

### Official Review · Reviewer_4LZz · 2023-07-06

**Soundness:** 3 good
**Presentation:** 3 good
**Contribution:** 3 good
**Rating:** 5
**Confidence:** 4

**Summary:**

In the paper, a novel method for view synthesis is proposed. More specifically, for given posed RGB input images, multi-resolution hash grid features are trained which are rendered to the image plane via volume rendering. The rendered feature is that passed through an MLP, processed with Spherical Harmonics (SH) feature encoding schemes, and then the final color is predicted. The main difference of the proposed system to previous NeRF models is that in 3D space, only feature grids without any MLP layers are optimized, and the view-dependent color prediction is only performed on the rendered features, i.e. "deferred shading" is applied.

**Strengths:**

- Results: I believe the main strengths of the paper is that the proposed system leads to good results, i.e. the proposed system outperforms SOTA methods such as Zip-NeRF or Instant-NGP on the two dataset types in both, the reported training times as well as the reported view synthesis quality.

- Ablation Study: A rather extensive ablation study is performed which I believe makes the manuscript stronger. It highlights the importance of the various components, and helps the reader to understand where the speed / quality improvements are coming from.

- Limitation Section: The manuscript contains a rather extensive limitation section which is beneficial as it portrays a more complete impression of the method.

- Manuscript organization: The manuscript is organized coherently and the information flow is rather easy to understand.

**Weaknesses:**

- Time comparison (L. 216, L. 229): The time comparison is, if I understood this correctly, not reported with running the methods on the same hardware, e.g. different GPUs (RTX 3090 vs V100) are used, and also times are simply adopted from other papers. As the time complexity is the main selling point of this work, I believe this should be reported more accurately by running the methods on the exact same hardware / setup.

- Qualitative Results - Animations: While the reported method achieves high-quality view synthesis performance measured by per-image metrics, the "smoothness" / "3d consistency" between frames is not shown. As the proposed system uses features instead of directly color predictions, the consistency between frames might be lower than for previous methods while not having a significant influence on the per-image metrics. It would there be very helpful to also show animations, e.g. in the form of fly-through videos, to compare methods with this focus.

- Unclear how the “pilot network” is used: Is simply the MLP size changed, or is this operating as the “default NeRF” models, i.e. per-point MLP calls are performed, and then color/density is aggregated via volume rendering? It would be helpful if the description of the pilot network would be made more precise.

- Figure 3: Unclear. What exactly is happening in the individual blocks? Is a per SH feature predicted, and then they are summed together? What is the “bottleneck arrow” indicating? Why is this skip connection actually necessary? The exact math formulations is also not contained in the respective Section 3.4. It would be helpful if the description of the deferred rendering part is more precise and clear formulas are used so that the reader can better understand the exact technique.

- Density prediction: How is the density obtained? Is this stored, similarly to the features, in a 3D hash grid?

- Qualitative Comparison - Figure 5: I think in Figure 5, also results from Zip-NeRF should be shown.

Typos / Unclear passages:

- L. 5: either in occupancy grids -> either in the form of occupancy grids
- L. 8: after neural network evaluation -> after neural network evaluations
- L. 19: View synthesis is a task that synthesizes -> For the task of view synthesis, unobserved views are synthesized …
- L. 36: near surface -> near the surface
- L. 39: standard volume rendering technique -> standard volume rendering techniques
- L. 58: NN runnings -> NN calls
- L. 61: to enable one sample on the surface -> to enable one sample-based rendering
- L. 69: modeling the specular effect -> modeling specular effects
- L. 96: represent F by use of MLPs. -> represent F by using MLPs
- L. 97: as N times MLP evaluations are required -> as N MLP evaluations are required
- L. 97/98: Alternatively, recent research shows that -> please add respective citation
- L. 100: organized in the forms of -> organized in the form of
- L. 110: the many times NN running is … -> the many NN calls are …
- L. 111: the importance of a large NN in realistic rendering -> the importance of a large NNs for realistic rendering
- L. 112 - 114: “Considering the fact that valuable samples on the surface are close in terms of spatial position, the queried feature vectors are also very similar as they are interpolated according to their position” -> unclear meaning
- L. 213: reports on the rendering -> reports the rendering
- L. 223: Using learning features organized in hash grids -> Using learned features organized in hash grids
- L. 249: Larger MLP (B, C) -> B does not indicate a “larger MLP” if I understood these symbols correctly. I would also suggest to structure the names and ablation study slightly different, e.g. with the typical “Baseline”, “+ GeLU”, “+larger MLP”, etc naming convention; currently, it is very cryptic and hard to understand.

Missing Citations:
- PlenOctrees For Real-time Rendering of Neural Radiance Fields, Yu et al., ICCV 2021
- Plenoxels Radiance Fields without Neural Networks, Yu et al., CVPR 2022

**Questions:**

- Are the time comparisons performed with respect to different hardware, e.g. RTX 3090 vs V100 GPUs?
- Could the authors expand on the "3D consistency" / "smoothness" of in-between frames when rendering a fly-through, compared to non-feature-based methods such as ZipNeRF?
- How exactly is the pilot network used? Is this predicting density and color per 3D sample point, or is this a smaller network but also operating on the rendered feature?
- Could the authors explain the SH-based deferred rendering technique more precise?
- Is the density predicted in 3D and stored in 3D hash grids, similar to the features?

**Limitations:**

- The authors discuss limitations of their work in the main manuscript. They mainly discuss the limitation of per-scene optimization and comparably slow / not yet real-time rendering times. I believe the limitation discussion is OK like it is; if some interesting failure mode could be illustrated with a figure, this could be interesting to add to the manuscript.

---

> ### Author Rebuttal · Authors · 2023-08-09
>
> Thanks for your valuable comments on our manuscript. We address your concerns as follows.
>
> * __Time comparison:__ In light of your suggestion, we report on the training times of various comparison methods measured on the same GPU RTX 3090. As shown in Tables 2 and 3, we can draw the same conclusion as in the original manuscript. Our method achieves a better rendering quality while taking less training time compared with the comparison methods. On the NeRF synthetic dataset, our VFR outperforms the SOTA NerfAcc by 1.51 dB in PSNR but using less training time, or achieves a similar PSNR as NerfAcc but 4+ times faster. On the real-world 360 dataset, our method offers a better rendering quality (29.48 dB) than NerfAcc (28.69 dB) while using 52 % of NerfAcc’s training time. Compared with the concurrent Zip-NeRF that achieves the SOTA PSNR results using 5.2 hours, our method deliveries a slightly better PSNR using only 11 minutes, which is 28 times faster than Zip-NeRF.
>
> Table 2. PSNR and training time on the NeRF synthetic dataset on one RTX 3090.
> |       | PSNR | Time
> | ------ | ------ | --- |
> | NeRF          | 31.69   | hours
> | Mip-NeRF   | 33.09   | hours
> | TensoRF      |  33.14  | 10 mins
> | Instant-NGP | 32.35  | 4.2 mins
> | NerfAcc       |  33.11  | 4.2 mins
> | Our VFR: 6K | 33.02 | 0.97 mins
> | Our VFR: 20K | __34.62__ | __3.3 mins__
>
> Table 3. PSNR and training time on the real-world 360 dataset on one RTX 3090.
> |       | PSNR | #Feature | Time
> | ------ | ------ | --- | --- |
> | NeRF          | 24.85   | N/A | days
> | Mip-NeRF   | 25.12   | N/A  |days
> | NeRF++      |  26.21  | N/A |days
> | Mip-NeRF360 | 29.11  | N/A |days
> | Instant-NGP       |  27.06  | 84M |0.81 hrs
> | Zip-NeRF | 29.82 | 84M | 5.2 hrs
> |NerfAcc | 28.69 | 34M | 11 mins
> |Our VFR: 20K| 29.48 | __34M__ | __5.7 mins__
> |Our VFR: 40K| __29.92__ | __34M__ | __11 mins__
>
> * __Qualitative Results - Animations:__ We do not observe the consistency difference between our rendered videos and those from the standard volume rendering. Both our VFR and the standard volume rendering method are based on global representations (representing a scene using a MLP and a multiresolution hash grid), which are recognized as the source of consistency between frames. Therefore, the rendered frames using our VFR are as smooth as those from the standard volume rendering. We will upload the rendered videos to demonstrate the frame consistency in the final version.
>
> * __Unclear how the “pilot network” is used:__ The pilot network is a small MLP operating as the “default NeRF” models, i.e., per-point MLP calls are performed, and then color/density is aggregated via volume rendering.
>
> * __Figure 3: Unclear:__ Thanks for pointing this out. For an integrated feature vector, the spatial MLP yields two feature vectors: SH feature vector and bottleneck feature vector. The SH feature vector will be spitted into small feature vectors for each SH coefficient, i.e.,  {$\mathbf{f}_0^0 | l=0$}, {$\mathbf{f}_1^{-1},  \mathbf{f}_1^{0}, \mathbf{f}_1^{1} | l=1$}, etc. In the SH feature encoding block, the SH feature vector $\mathbf{f}_m^l$ will be multiplied with SH coefficient (basis function) $Y_l^m(\mathbf{d})$ with view direction $\mathbf{d}$ to form one SH feature encoding vector $\mathbf{e}_m^l$, written as $\mathbf{e}_m^l = \mathbf{f}_m^l Y_l^m(\mathbf{d})$. A comprehensive SH encoding is derived by concatenating all encoding vectors as $\mathbf{E}$ = {$\mathbf{e}_0^0, \mathbf{e}_1^{-1}, \mathbf{e}_1^{0} ...$}, which will be further concatenated with the bottleneck feature vector (view direction independent). This concatenated vector is used to predict the final rendered color by the directional MLP. Based on the above description, we summarize answers to your questions as follows:
>
>     * For each SH coefficient $Y_l^m(\mathbf{d})$, the spatial MLP will predict a small SH feature vector $\mathbf{f}_m^l$ for it. The SH feature encodings $\mathbf{e}_m^l$ will be concatenated to form a comprehensive encoding.
>     * The bottleneck arrow indicates a bottleneck feature vector predicted by the spatial MLP.
>     * The skip connection is necessary because we want the bottleneck feature vector to be independent to view direction for the prediction of diffuse color.
>
>     We will add these details in the final version as suggested by the reviewer.
>
> * __Density prediction:__ The density is obtained by applying a linear layer to the queried feature vector from the underlying multi-resolution hash encoding. As this linear layer only has one output neuron, it has a negligible effect on the overall training and rendering times.
>
> * __Qualitative Comparison - Figure 5:__ The results from Zip-NeRF are similar to ours in visual. We will add the results from Zip-NeRF in Figure 5 as per your suggestion.
>
> * __Typos / Unclear passages:__ Thanks for pointing out these typos / unclear passages. We will implement these corrections in our final paper. For line 249, we apologize for the confusion caused by this small mistake. The correct description should be “Larger MLP (C) does increase … ”. We will revise the description of ablation study by following your suggestion in structuring the names and ablation study using the typical “Baseline”, “+ GeLU”, “+larger MLP”.
>
> * __Missing Citations:__ We will include these citations in our final version of the paper.
>
> * __Limitations--interesting failure mode:__ We have not found other failure modes related to our method in addition to the convergence issue (solved by our proposed pilot network). One potential limitation of our method is that it may not work well for semitransparent objects as our method integrates feature vectors first and then predicts a single final color. This property is not well reflected in the NeRF synthetic and real-world 360 datasets and requires further investigation in our future work. We will add this discussion to our final version.

---

> > ### Comment · Reviewer_4LZz · 2023-08-14
> >
> > I thank the authors very much for this extensive and informative rebuttal. I do not have further questions from my side.

---

### Author Rebuttal · Authors · 2023-08-10

We would like to express our sincere gratitude to all the reviewers for their precious time and thoughtful review of this manuscript. The comments and suggestions raised are extremely valuable and constructive, and very helpful for improving the quality of the manuscript.

Please see the detailed response to each reviewer in the separate response.

---

### Decision · Program_Chairs · 2023-09-21

**Decision:**

Accept (poster)

**Comment:**

As discussed in the rebuttal and post-rebuttal discussion, feature integration is not an entirely new idea but has been explored to some extent in generative 3D methods, such as StyleNeRF and EG3D. The authors should edit their discussion regarding novelty and positioning to prior work. Many other suggestions were made by the reviewers to improve the manuscript - please incorporate these in your camera-ready paper.